# An Environmentally Sensitive Silk Fibroin/Chitosan Hydrogel and Its Drug Release Behaviors

**DOI:** 10.3390/polym11121980

**Published:** 2019-12-01

**Authors:** Zhangpeng Xu, Erni Tang, Huijing Zhao

**Affiliations:** 1National Engineering Laboratory for Modern Silk, College of Textile and Clothing Engineering, Soochow University, No. 199 Ren’ai Road, Industrial Park, Suzhou 215123, China; 20184215011@stu.suda.edu.cn; 2China Leather and Footwear Industry Research Institute (Jinjiang) Co., Ltd. No., 150 Wenhua Road, Hongshan Comprehensive District, Jinjiang 362200, China; tangerni@foxmail.com; 3Nantong Textile & Silk Industrial Technology Research Institute, Building D1, No 266 Xinshiji Ave, Jianghai Intellectual Park, Tongzhou, Nantong 226001, China

**Keywords:** hydrogel, silk fibroin, chitosan, stimuli responsive swelling performance, sustained drug release

## Abstract

To fabricate environmentally sensitive hydrogels with better biocompatibility, natural materials such as protein and polysaccharide have been widely used. Environmentally sensitive hydrogels can be used as a drug carrier for sustained drug release due to its stimulus responsive performance. The relationship between the internal structure of hydrogels and their drug delivery behaviors remains indeterminate. In this study, environmentally sensitive hydrogels fabricated by blending silk fibroin/chitosan with different mass ratios were successfully prepared using 1-(3-Dimethylaminopropyl)-3-ethylcarbodiimide (EDC)/N-Hydroxysuccinimide (NHS) cross-linking agent. Scanning-electron microscopy (SEM) images showed the microcosmic surface of the gel had a 3-D network-like and interconnected pore structure. The N_2_ adsorption–desorption method disclosed the existence of macroporous and mesoporous structures in the internal structure of hydrogels. Data of compression tests showed its good mechanical performance. The swelling performance of hydrogels exhibited stimuli responsiveness at different pH and ion concentration. With the increase of pH and ion concentration, the swelling ratios of hydrogels (silk fibroin (SF)/ chitosan (CS) = 8/2 and 7/3) decreased. Methylene blue (MB) was loaded into the hydrogels to confirm the potential of sustained drug release and pH-responsive behavior. Therefore, due to the porous structure, stable mechanical strength, stimuli responsive swelling performance, and drug release behaviors, the SF/CS composite hydrogels have potential applications in controlled drug release.

## 1. Introduction

During medical treatment and care, oral administration is always convenient and efficient. It can avoid many clinical problems caused by injection, thus alleviating patients’ pain. However, the disadvantage is that the absorption of drugs is slow and irregular, and some drugs will be easily destroyed by the gastric juice [1]. As an alternative, drugs can be loaded into biocompatible materials and applied to the human body. The drug release process can last for a long time to ensure the effectiveness of the treatment [2].

Hydrogel is a kind of hydrophilic, 3-D network polymer material. It is formed by physical or chemical cross-linking [3,4,5]. Physical cross-links are reversible bonds between linear polymers generated with light, heat, and other radiation, or formed by hydrogen bonds, ionic bonds, and other interactions. Chemical cross-linking is the covalent bond between polymers, thus the internal structure of hydrogels prepared by chemical cross-linking is relatively stable. The physical properties of hydrogel are similar to the properties of living tissues, such as high water content, rubbery texture, and low interfacial tension, which contribute to its outstanding biocompatibility with the surrounding tissues [6]. The mechanical property, shape, and swelling performance of some hydrogels tend to change with the external conditions, such as temperature, pH value, and ion concentration [7,8,9]. Drugs can be delivered through hydrogels with stimuli responsiveness to pH, magnetism, ultrasound, temperature changes, and electrical effects [8]. The stimuli responsiveness of hydrogel makes it potential material for drug release, which has been proved in previous studies [10,11,12]. Cross-linking mechanism or pore structure of hydrogels may be the cause of stimuli responsiveness, which has not been revealed. In this study, a pH- and ion intensity stimuli-responsive silk fibroin/chitosan hydrogel was prepared by chemical cross-linking. The mechanism of stimulus responsiveness of the hydrogel was also indicated.

Silk fibroin (SF) is a kind of natural material with high molecular weight, good biocompatibility, and mechanical strength [13]. Hydrogel made of silk fibroin has broad application prospects in artificial skin, wound dressing, sustained drug release, tissue engineering scaffolds, and so on [14,15,16,17]. Chitosan (CS) is a natural basic polysaccharide polymer, which exhibits antimicrobial properties. Hydrogel made of chitosan has been used for wound dressings and drug carriers [18]. More importantly, chitosan-based hydrogel has been proved to be stimuli responsive [19,20,21], which is an ideal material for our research theme. The composite hydrogel consists of SF and CS was cross-linked by 1-(3-Dimethylaminopropyl)-3-ethylcarbodiimide (EDC)/N-Hydroxysuccinimide (NHS) to form amino bonds between the molecules. In order to determine if the SF/CS composite hydrogel has ideal physical properties and stimuli responsiveness for controlled drug release, some characterizations have been conducted, including mechanical strength, swelling performance, and drug release behaviors. Furthermore, SEM, N_2_ adsorption—desorption, FTIR, XRD, thermogravimetric properties, and rheological behavior were also investigated to study the relations between structure and properties.

## 2. Materials and Methods

### 2.1. Chemicals and Reagents Used

The *Bombyx mori* silkworm silk was purchased from Huzhou in Zhejiang province, China. Chitosan (Molecular weight =190~310 kDa, >75% deacetylated) and EDC/NHS were purchased from Sigma-Aldrich Chemical Co. (St. Louis, MO, USA). The ethanol, sodium hydroxide, hydrochloric acid, phosphate buffer saline (PBS), and anhydrous sodium carbonate were all of analytical grade and purchased from Sigma-Aldrich Chemical Co. (St. Louis, MO, USA).

### 2.2. Preparation of Silk Fibroin/Chitosan Hydrogel

#### 2.2.1. Preparation of Silk Fibroin Solution

The process of preparing silk fibroin solution was described previously [14,22]. Sodium carbonate was added into boiled deionized water, as well as the silkworm raw silk. The mixture was stirred for 30 min to remove the sericin. The treated silk was then washed five times with lukewarm deionized water to remove residual sericin. After drying, the extracted SF was added to 9.3 M LiBr solution at a bath ratio of 2.7:10 and incubated for 4–6 h at 60 °C. The dissolved solution was loaded into dialysis bags (MWCO 3500 Da) and dialyzed for 3 d in deionized water. After dialysis, the SF solution was centrifuged at 9000 rpm for 20 min at 4 °C to obtain supernatant, the pure SF solution. The pure SF solution was kept in the refrigerator at 4 °C until use.

#### 2.2.2. Preparation of Chitosan Solution

Chitosan solution with 2% mass fraction was prepared in 1% acetic acid and stirred on a hot plate for 2 h to facilitate its dissolution.

#### 2.2.3. Preparation of the Composite Hydrogel

Silk fibroin aqueous solution with 3% mass fraction and chitosan solution with 2% mass fraction were mixed in mass ratios of 10/0, 9/1, 8/2, 7/3, and 6/4, stirred evenly in a water bath at 37 °C. EDC and NHS (M/M = 2:1) were mixed in 95% ethanol. The weight of EDC accounted for 20% of silk fibroin. The SF/CS mixture was transferred into centrifuge tubes and centrifuged for 15 min at 3000 r/min for defoaming. The centrifuge tubes were kept in a biochemical incubator at 37 °C for 48 h to form hydrogels. The obtained SF/CS hydrogels were placed in deionized water for 24 h to remove residual acetic acid, ethanol, and the free, unreacted EDC-NHS. The deionized water was changed every 2 h. For a comparison purpose, 3% SF solution was kept at 37 °C for 7 d to form pure SF hydrogel (NSF).

### 2.3. Characterization of the Hydrogel

#### 2.3.1. Scanning Electron Microscope (SEM)

In characterization studies, the freeze-dried procedure was used to turn the hydrogels with mass ratios of 10/0, 9/1, 8/2, 7/3, and 6/4 into xerogels. The hydrogels were frozen rapidly by liquid nitrogen and then dried in vacuum to preserve the internal structure. The morphology of the xerogels was analyzed using SEM (S-4800, Hitachi, Japan) after being coated with gold for 90 s.

#### 2.3.2. N_2_ Adsorption-Desorption Experiment

The pore volume and pore size distribution of the hydrogels at mass ratios of 8/2 and 7/3 were analyzed by nitrogen adsorption-desorption at 77 K, determined by the BJH method with the ASAP 2020 surface area analyzer (Micromeritics Instrument Corp, Norcross, GA, USA). The specific surface area was based on the Brunauer–Emmett–Teller (BET) method [23].

#### 2.3.3. Fourier Transform Infrared Spectroscopy (FTIR)

To clarify the chemical structure, FTIR (Thermo Fisher Scientific, Waltham, MA, USA) spectra of the xerogels with mass ratios of 10/0, 9/1, 8/2, 7/3, and 6/4 were recorded from 400 to 4000 cm^−1^ frequency range.

#### 2.3.4. X-ray Diffraction (XRD)

The xerogel samples were measured by X-ray diffraction (X’Pert Pro MPD, PANalytical BV, Almelo, The Netherlands). The scanning speed was 6°·min^–1^ and the recorded region of 2θ was 5° to 45°.

#### 2.3.5. Thermogravimetric Analysis (TGA)

Thermogravimetric analysis of all freeze-dried hydrogels, NSF, and pure CS was carried out in a Perkin Elmer instrument (Diamond 5700, Waltham, MA, USA), in the presence of nitrogen atmosphere; all samples were heated at the scanning rate of 10 °C·min^–1^ for the range from 20 to 600 °C.

#### 2.3.6. Rheological Studies

Rheological properties of hydrogels were tested using AR-2000 rheometer (TA Instrument, New Castle, USA) fitted with parallel plate (diameter of 20 mm) kept at a gap distance of 1 mm. The samples were pretreated to fit the parallel plate. Strain sweep analysis for the range from 0% to 60% was conducted to show the viscoelastic behavior of the hydrogels. Rheological analysis was performed in linear viscoelastic region at 37 °C.

### 2.4. Mechanical Properties

The stress–strain curves in compression were obtained using Instron-3365 material testing machine (Instron, Boston, MA, USA). Hydrogels with mass ratios of 10/0, 9/1, 8/2, 7/3, and 6/4 were cut into cylinders (diameter of 14 mm, height of 12 mm). All samples were tested at a compression speed of 5 mm·min^–1^.

The compression-resilience property was tested using a TMS-PRO texture analyzer (Food Techonology Corporation, Sterling, VA, USA). Hydrogels with mass ratios of 10/0, 9/1, 8/2, 7/3, and 6/4 were cut into cylinders (diameter of 12 mm, height of 10 mm). The maximum compression displacement was 40% of the original height. The compression-resilience rate was then calculated on the basis of the obtained curves.

### 2.5. Swelling Performance and Stimulus Responsive Behaviors

To test the swelling performance of hydrogels, following method was implemented. A weighed amount (40 mg) of each hydrogel was taken in a beaker containing measured amount (200 mL) of PBS solvent. At a given time, the weight of swollen hydrogel was determined by removing extra solvent by absorbing at surface. It was then soaked in PBS solvent to achieve equilibrium of swelling. Equations (1) and (2) were used to calculate the swelling ratio. “*W_t_*” was the weight of swollen gel at time “*t*”, “*W*_1_” was the weight of dry hydrogel and “*W*_2_” was a weight of swollen gel that reached equilibrium.
(1)$$\mathrm{Dynamic}\;\mathrm{swelling}\;\mathrm{ratio}=\frac{W_{t}-W_{1}}{W_{1}}\times 100\%$$
(2)$$\mathrm{Equilibrium}\;\mathrm{swelling}\;\mathrm{ratio}=\frac{W_{2}-W_{1}}{W_{1}}\times 100\%$$

The hydrogels with mass ratios of 8/2 and 7/3 were chosen to determine the relationship between swelling performance and environmental conditions such as temperature, ion concentration, and pH. The testing procedures were consistent with the approaches above. Hydrogels were soaked in PBS solutions at different temperature (20 °C, 37 °C, 60 °C) and pH (2.2, 7.4, 9.0), as well as in NaCl solutions at the concentration of 0.6 M, 1.5 M, 2.5 M, and 5.0 M.

### 2.6. Drug Loading and In Vitro Release

Dry hydrogel of 40 mg was soaked in 15 mL MB solution (2 mg mL^−1^) to a constant weight to load drugs. Equation (3) was used to calculate the drug loading rate. “*m_t_*” was the weight of dry hydrogel after loading drugs and “*m*_0_” was the weight of dry gel.
(3)$$\mathrm{Drug}\;\mathrm{loading}\;\mathrm{rate}=\frac{m_{t}-m_{0}}{m_{0}}\times 100\%$$

Dry hydrogels with mass ratios of 8/2 and 7/3 loaded with MB were placed into 50 mL PBS (pH value = 7.4, 2.2) at constant temperature (37 °C) with shaking at 50 rpm. At time intervals, 2 mL of release solution was withdrawn, and 2 mL of PBS solution was added to maintain constant volume. The concentrations of the drug released from hydrogels were analyzed by the UV-visible spectrophotometric. Standard curves of MB in PBS were obtained (Appendix A). The position of the absorption peak of MB was 664 nm (Appendix A). Equation (4) was used to calculate the drug loading rate. “*V*” is the volume of PBS, “*c_n_*” is the concentration of MB in PBS equivalent to absorbances, “*V_d_*” is the volume of PBS withdrawn, and “*m_L_*”is the load amount of MB.
(4)$$\mathrm{Drug}\;\mathrm{release}\;\mathrm{rate}=\frac{V_{cn}+V_{d}\sum_{1}^{n-1}c_{i}}{m_{L}}\times 100\%$$

### 2.7. Statistical Analysis

Results are presented as mean ± SD. The mean value was calculated according to at least 3 tests. Comparison of the mean values of the datasets was assessed by one-way analysis of variance (ANOVA).

## 3. Result and Discussion

### 3.1. Preparation of Silk Fibroin/Chitosan Hydrogels

In the pre-experiments, SF/CS composite hydrogel was successfully prepared. Although the hydrogel formed by chitosan alone has many biomedical advantages and benign gelation condition, it has been proved that the addition of silk fibroin can improve the mechanical properties of hydrogels [24]. The SF/CS hydrogels (at mass ratios of 10/0, 9/1, 8/2, 7/3 and 6/4) were prepared by mixing SF, CS solution together, and EDC/NHS played a role as the cross-linker to make carboxyl groups from SF and amino groups from CS formed amide bond (Figure 1A). The amide bond in the hydrogel network made the internal structure more compact [25]. The schematic representation of the composite hydrogel is shown in Figure 1B. Compared with composite hydrogels, the structure of the NSF hydrogel was instable and could not meet the basic requirement for mechanical test (Figure 1C). Compared with the other composite hydrogels, hydrogels with mass ratios of 8/2 and 7/3 had better molding performance, thus suitable for drug loading and other tests.

### 3.2. Characterization of the Hydrogel

#### 3.2.1. SEM

In the freeze-dried procedure, hydrogels were frozen rapidly by liquid nitrogen, then water in the internal space turned into ice immediately and sublimed in vacuum to preserve the internal structure of hydrogels [26]. SEM images of NSF hydrogel and composite hydrogels are shown in Figure 2. The pore structure of NSF hydrogel was irregular and incomplete, while the inner pore distribution of composite hydrogels was regular, and the pore structure presented a 3-D network structure similar to hive. This is due to the crosslinking density in the hydrogel, which is associated with the content of SF or CS. When the content of CS was very low (SF/CS = 10/0, 9/1, and 8/2), the pore size of these hydrogels presented as larger than hydrogels (SF/CS = 7/3, 6/4). When the content of CS was higher (SF/CS = 7/3, 6/4), it can be clearly seen that hydrogels had smaller pore size and denser pore structure than hydrogels (SF/CS = 9/1, 8/2). During the formation of hydrogels, with the addition of CS, the mixed solution had higher viscosity than pure SF solution (Appendix A) and the flexibility of molecule chains was weakened. The loss of hydrogen bond and the enhancement of hydrophobic force made it more possible for molecule chains to aggregate together to form hydrogels with smaller pore size. The surface of hydrogels (SF/CS = 7/3, 6/4) had a 3-D network-like and interconnected pore structure (Figure 2), which indicates that the hydrogel has a potential application for drug loading and release [27].

#### 3.2.2. Analysis of the Pore Structure

In view of the ultimate application as scaffolds for delivering drugs or housing fluids from the surrounding tissues, porous structure is essential. The N_2_ adsorption–desorption isotherms and pore size distributions of hydrogels at mass ratios of 8/2 and 7/3 are shown in Figure 3. The isotherms of hydrogels exhibited a type-II curve according to the IUPAC classification (macroporous: Pore diameter > 50 nm; mesoporus: Pore diameter between 2–50 nm; microporus: Pore diameter < 2 nm), which indicated the macroporous structure. The specific surface area was calculated to be 12.53 m^2^/g (SF/CS = 8/2) and 9.62 m^2^/g (SF/CS = 7/3) based on the BET method. The 8/2 hydrogel had larger specific surface area than that of the 7/3 hydrogel. Moreover, the BJH method for the pore size distribution showed that the pore size was centralized at approximately 15.45 nm (SF/CS = 8/2) and 11.27 nm (SF/CS = 7/3), indicating the existence of mesopores. The results demonstrated that macroporous and mesoporous structures existed in the internal structure of hydrogels simultaneously.

#### 3.2.3. FTIR

The chemical structures of SF, CS, and SF/CS hydrogels were confirmed by FTIR (Figure 4). The broad absorption peak at 3423 cm^−1^ was assigned to the hydroxyl and amino groups of CS. The anti-symmetrical contraction vibration of C-O-C group at 1158 cm^−1^ and 894 cm^−1^ represented the polysaccharide structure of CS. The absorption peaks of NSF (curve a in Figure 4) appeared at 1659 cm^−1^ and 1629 cm^−1^ (amide I), 1532 cm^−1^ (amide II), 1234 cm^−1^ (amide III), and 700 cm^−1^ (amide V), indicating the α-helix, β-sheet, and random coil of SF. The absorption peaks of silk fibroin (curve b in Figure 4) appears at 1629 cm^−1^ (amide I), 1532 cm^−1^ (amide II), indicating that β-sheet was the main structure of the cross-linked hydrogels. Compared with the cross-linked SF hydrogel (curves b in Figure 4), with the addition of CS, peak of the carboxyl group of SF at 948 cm^−1^ disappeared, and a new peak, the polysaccharide structure of CS, appeared at 894 cm^−1^ (curve c to f in Figure 4). It was due to the new amide bond formed by carboxyl group of SF and amino group of CS [28].

#### 3.2.4. XRD

X-ray diffraction (XRD) was applied to analyze the condensed state structure of silk fibroin, including three molecular chain conformation (silk I, silk II, and random coil) and two crystal type (α-helix and β-sheet) [29]. There was a diffraction peak near 21.7°and 8.7° in CS (curve g in Figure 5), indicating CS had some crystalline areas and its molecular structure was dominated by amorphous structure [30]. The curve of NSF (curve a in Figure 5) had a sharp diffraction peak at 20.7° (silk II) and a weak peak near 24.5° (silk I), which meant NSF had the crystalline structure of silk II. With the addition of chitosan, the peak at 24.5° (silk I) gradually disappeared, while the diffraction peak at 20.4° (silk II) still existed. Hence, silk II (β-sheet) was the main structure in the SF/CS hydrogels, and it was consistent with the results of FTIR.

#### 3.2.5. TGA

The thermograms of SF, CS, and composite hydrogels were shown in Appendix A, which confirmed that the hydrogels were thermally more stable due to the formation of crosslinking. SF started to decompose at 215.74 °C and the apparent weight loss temperature was 297.72 °C, which was mainly due to the heat absorption caused by SF decomposition [31,32]. Compared with SF, NSF showed a higher decomposition temperature at 301.39 °C as a result of the stable β-sheet structure in NSF [33]. CS started to decompose at 242.5 °C, and the apparent weight loss temperature was 288.95, which was mainly due to the dehydration of sugar ring, deacetylation of chitosan, depolymerization, and decomposition of deacetylated structures [34]. The decomposition temperatures of SF/CS composite hydrogels with mass ratios of 10/0, 9/1, 8/2, 7/3, and 6/4 were higher than NSF, because hydrogels formed a condensed network by crosslinking between SF and CS [35].

#### 3.2.6. Rheological Properties

The rheological properties of hydrogels at mass ratios of 9/1, 8/2, 7/3 and 6/4 were explored by detecting the storage modulus (G′) and the loss modulus (G′′) of the hydrogels (Figure 6). When the content of silk fibroin in the composite gel was high, the hydrogel exhibited the largest G’, which meant the addition of SF could enhance the viscoelasticity of the hydrogels. This was mainly because of the effects of interlocking entangled chains and hydrogen bond between SF and CS [36]. The curves of G′ and G′′ intersected when the strain was very small (10%~20%), and then G′ < G′′, the hydrogels turned from viscoelastic state into viscous flow state due to the destruction of the internal network of hydrogels at high strain.

### 3.3. Mechanical Property

The results of the compressive tests showed the SF/CS composite hydrogels prepared by chemical crosslinking had good compressive strength and recovery property. The stress–strain curves (strain ranged from 0% to 60%) of the composite hydrogels were shown in Figure 7A. The hydrogel with mass ratio of 9/1 exhibited different stress–strain curve shape due to its bad molding performance. The 8/2 presented the best compressive properties while the 6/4 showed the worst. The hydrogen bonds and hydrophobic interactions between SF and CS were enhanced by crosslinking, which also enhanced the stability of the internal network structure [37]. Therefore, as the content of CS increased, the compressive strength of the composite hydrogels gradually increased. When the content of SF increased, the mixed solution had lower viscosity and larger fluidity, and the formed skeleton of hydrogels became loose. In Figure 7B, comparisons of the recovery properties were carried out. The hydrogels at mass ratios of 8/2, 7/3, and 6/4 exhibited higher resilience rate (>90%) than 9/1, which was due to their high crosslinking density. The formation of the network structure between chains of SF and CS can improve the elasticity of hydrogels [38]. Hence, the SF/CS composite hydrogels prepared by chemical cross-linking have stable compressive strength and good elasticity.

### 3.4. Swelling Performance and Stimulus Responsive Behaviors

The dynamic swelling ratio (SR) was tested using deionized water at 37 °C and equilibrium swelling ratio (ESR) of hydrogels was determined in PBS (pH = 7.4, 37 °C). The swelling behavior reached equilibrium after 50 h. The dynamic swelling ratio (SR) (Figure 8A) and equilibrium swelling ratio (ESR) (Figure 8B) increased with the content of CS. At the beginning of swelling process, water was absorbed by capillary existed in the internal structure of hydrogels. The 3-D network, porous structure provided channels for molecules to enter and exit. The hydrophilic groups (-OH/-COOH/-COO-) were combined with water molecules by coordination bond or hydrogen bond to form a hydration layer. Both SF and CS have hydrophilic groups on their molecular chains, which is beneficial for the formation of hydration layers. SF is a relatively hydrophobic protein. As its content increased, the hydrophilicity of the matrices decreased. Moreover, the channels formed inside hydrogels with large pores can make the gels swelling and deswelling by water convection [39]. In previous studies, the carrageenan-based hydrogel showed a great swelling ratio (140%~299%) [40]. The equilibrium swelling ratio of silk fibroin/polyurethane composite hydrogel was about 5% [41]. The swelling ratio of a chemically modified chitosan biopolymer is 50%~175% [42]. Compared with the related research, the swelling ratio of SF/CS composite hydrogel prepared in this study was relatively high.

The stimulus responsiveness of swelling performance was determined by comparing the swelling behaviors of SF/CS composite hydrogels with mass ratios of 8/2 and 7/3 at different temperatures, pH, and ionic concentrations. Temperature (37 °C), pH (2.2, 7.4), and low ionic concentration of the swelling medium were applied to simulate the physiological environment of human body. The hydrogel with higher crosslinking density attained more compact structure, which decreased the ESR [43]. The ESR of the SF/CS composite hydrogel (Figure 8C) has no linear relationship with changes in temperature. In previous studies, SF or CS temperature-responsive hydrogels were formed by modifying the structure of SF/CS or blending with other temperature-responsive materials [41,44]. The ESR of SF/CS composite hydrogel (Figure 8D) decreased remarkably with the raise of the concentration of NaCl. As the swelling medium, NaCl solution contains a large number of free ions, which increased the osmotic pressure of the swelling medium and minimized the inward movement of solvent solution [45]. After the combination of hydrogel with hydrated cations, electrostatic repulsion force generated between hydrogel and free ions, which made the structure of hydrogel network shrink. A decrease in ESR was observed when the pH value of PBS increased (Figure 8E), which proved the pH stimuli responsiveness of the SF/CS hydrogels. Both SF and CS had positive charges when the pH was 1~3 [46], which made the molecular chains of hydrogels generate electrostatic repulsion and led to the higher ESR. When the pH was 4~14, SF had negative charges instead and exhibited electrostatic attraction with CS. Hence, the internal structure of hydrogels was compact and uneasy to swell.

### 3.5. Drug Loading and In Vitro Release

The in vitro drug release behaviors of SF/CS hydrogels at mass ratios of 8/2 and 7/3 in PBS as release medium were studied. In Figure 9A, the drug loading rate of the 7/3 hydrogel (>50%) was obviously higher than that of the 8/2 hydrogel (20%). In light of the way to load drug, the drug loading rate is largely related to the swelling performance. As is known in the analysis of swelling performance, the 7/3 hydrogel has higher ESR, so that it has more space in the internal structure of hydrogel after swelling and can contain more drug solution.

In the earlier release period (0~10 h), the hydrogels showed higher release rate (Figure 9B,C). The release stage reached equilibrium after 35 h. Compared with normal physiological environment (PBS, pH = 7.4, 37 °C), hydrogels in an acidic environment (PBS, pH = 2.2, 37 °C) had more drug release amount in total. It also confirmed the swelling performance of the SF/CS hydrogels at different pH. In previous studies, a citric acid crosslinked NaCMC-HPMC hydrogel reached equilibrium after 12 h and release rate of methylene blue was approximately 62.75 ± 1.40% within 4 h [47]. The pH-responsive hydrogel prepared by gelatin and bacterial cellulose reached equilibrium after 10 h, the release rate was 60% in an acidic environment and 80% in a normal physiological environment [48]. Hence, the SF/CS composite hydrogels prepared in this study can reduce the burst release and maintain sustained release for a long time. It has the potential for smart drug release.

## 4. Conclusions

A series of stimuli responsive composite hydrogels were successfully prepared. The composite hydrogels were prepared based on SF and CS, cross-linked by EDC/NHS. These hydrogels exhibited porous structures in the internal structure, including macropores and mesopores. The composite hydrogel appeared as an elastomer with excellent mechanical compressive property. The degree of swelling was positively related to the content of CS. The swelling performance of the hydrogels at different pH and ion concentrations showed stimuli responsiveness. With the increase of pH and ion concentration, the swelling ratios of hydrogels (SF/CS = 8/2 and 7/3) decrease. The in vitro drug release rate also had a pH-responsive behavior, which suggested the potential for controlled drug release. The SF/CS composite hydrogel prepared in this study is a prospective material for sustained drug release.

## Figures and Tables

**Figure 1 polymers-11-01980-f001:**
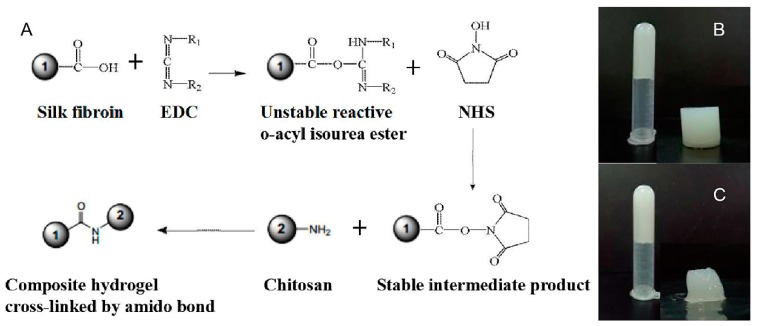
Synthesis of silk fibroin (SF)/chitosan (CS) hydrogels. (**A**) Cross-linking mechanism, (**B**) schematic representation of SF/CS hydrogel with mass ratio of 7/3, (**C**) schematic representation of NSF.

**Figure 2 polymers-11-01980-f002:**
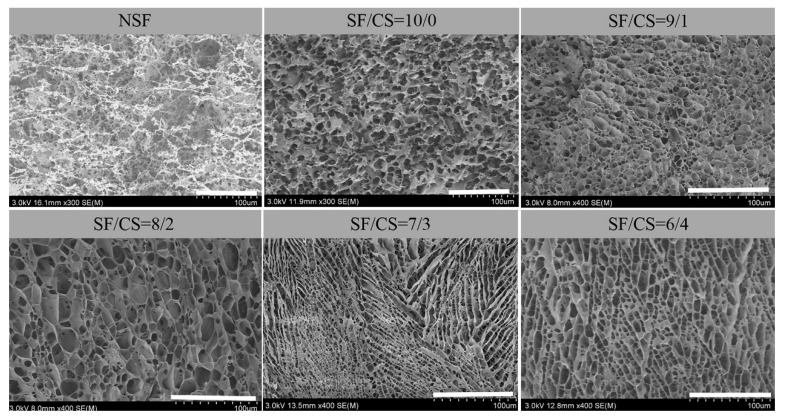
SEM images of hydrogels. Scale bar: 100 μm.

**Figure 3 polymers-11-01980-f003:**
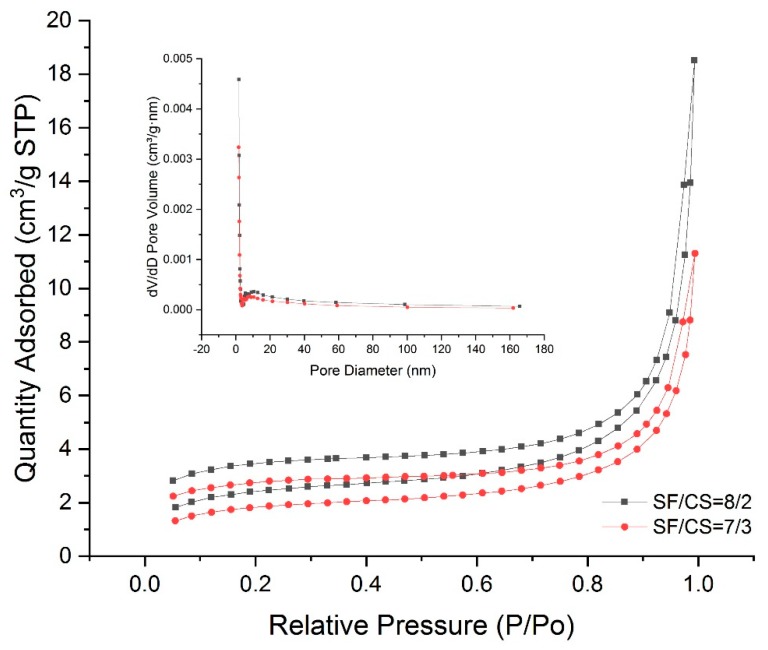
N_2_ adsorption–desorption isotherms and pore size distribution of hydrogels at mass ratios of 8/2 and 7/3.

**Figure 4 polymers-11-01980-f004:**
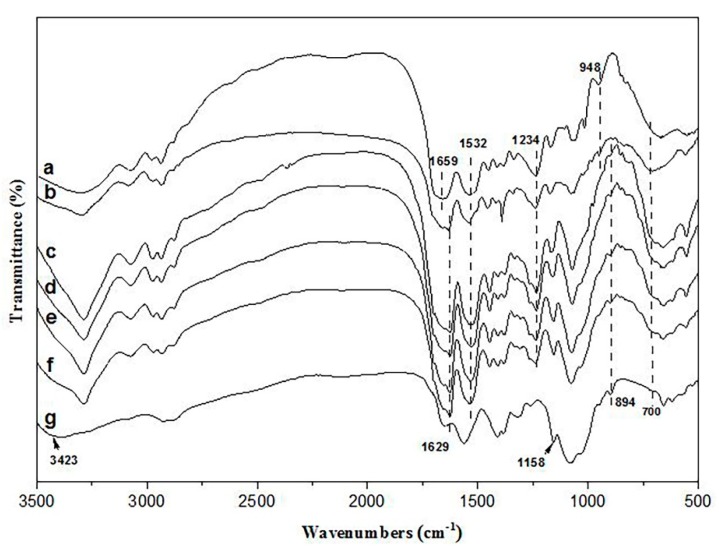
FTIR spectra. (**a**) NSF, (**b**) SF/CS = 10/0, (**c**) SF/CS = 9/1, (**d**) SF/CS = 8/2, € SF/CS = 7/3, (**f**) SF/CS = 6/4, (**g**) chitosan.

**Figure 5 polymers-11-01980-f005:**
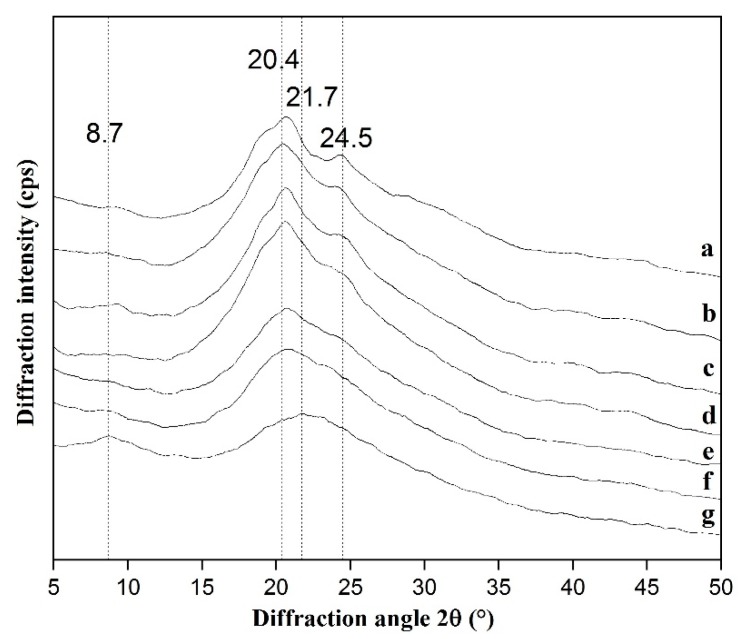
XRD diagram. (**a**) NSF, (**b**) SF/CS = 10/0, (**c**) SF/CS = 9/1, (**d**) SF/CS = 8/2, € SF/CS = 7/3, (**f**) SF/CS = 6/4, (**g**) chitosan.

**Figure 6 polymers-11-01980-f006:**
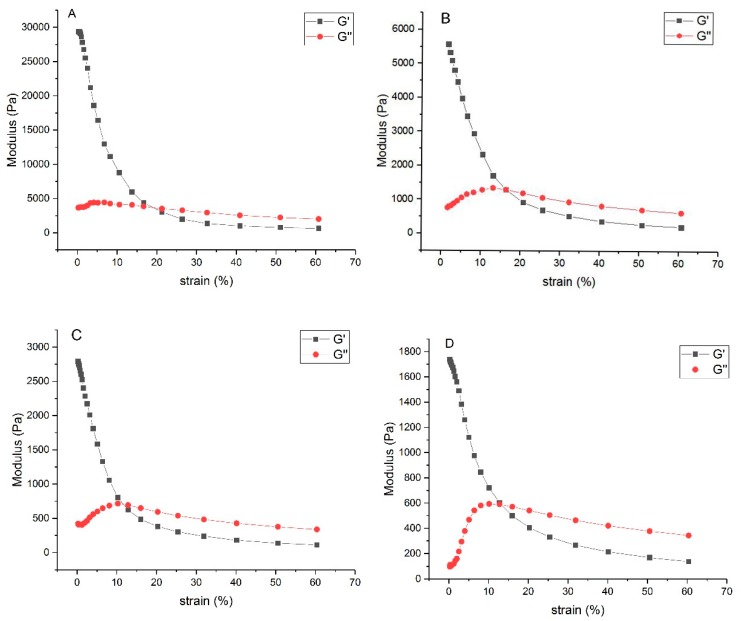
Rheological properties of hydrogels. (**A**) SF/CS = 9/1, (**B**) SF/CS = 8/2, (**C**) SF/CS = 7/3, and (**D**) SF/CS = 6/4.

**Figure 7 polymers-11-01980-f007:**
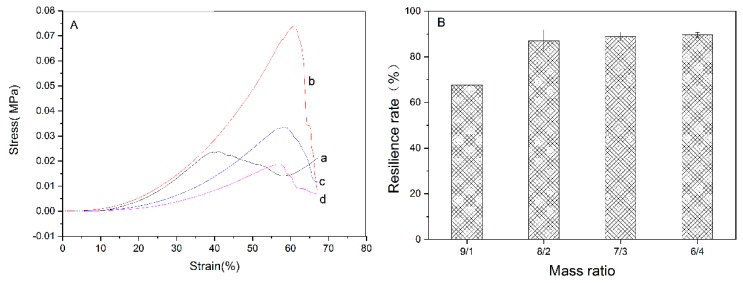
Mechanical properties. (**A**) Stress–strain curves. (**a**) 9/1, (**b**) 8/2, (**c**) 7/3, and (**d**) 6/4, (**B**) resilience rate.

**Figure 8 polymers-11-01980-f008:**
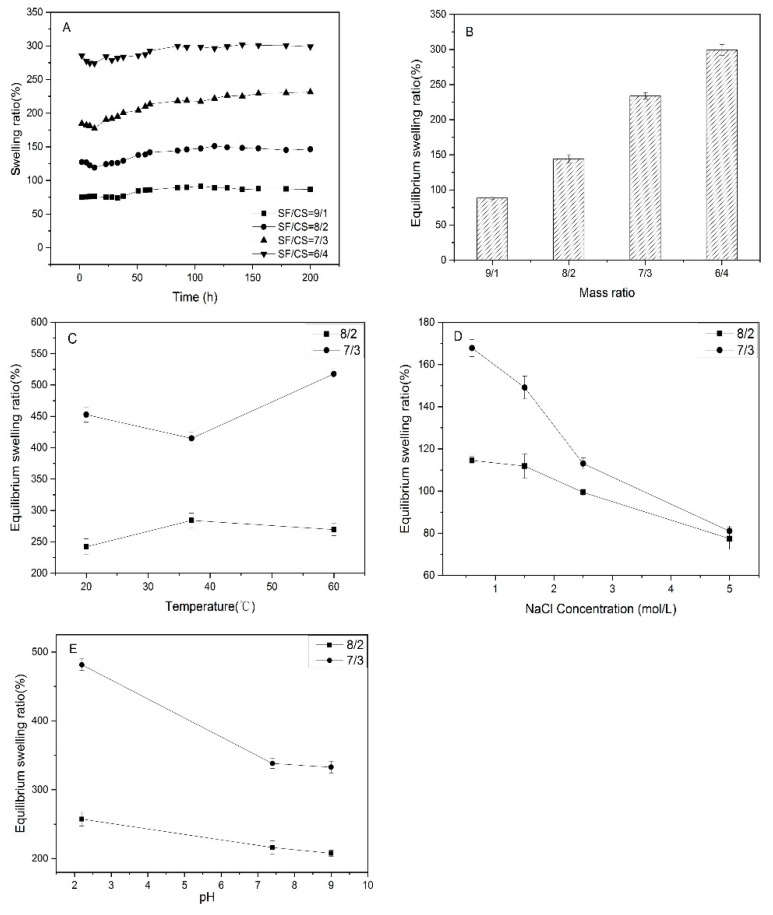
Swelling performance and swelling sensitivity. (**A**) Dynamic swelling ratio; (**B**) equilibrium swelling ratio; (**C**) equilibrium swelling ratio at 20 °C, 37 °C, and 60 °C; (**D**) equilibrium swelling ratio at different ionic concentrations: 0.6 M, 1.5 M, 2.5 M, and 5.0 M; (**E**) equilibrium swelling ratio at pH = 2.2, 7.4, and 9.0.

**Figure 9 polymers-11-01980-f009:**
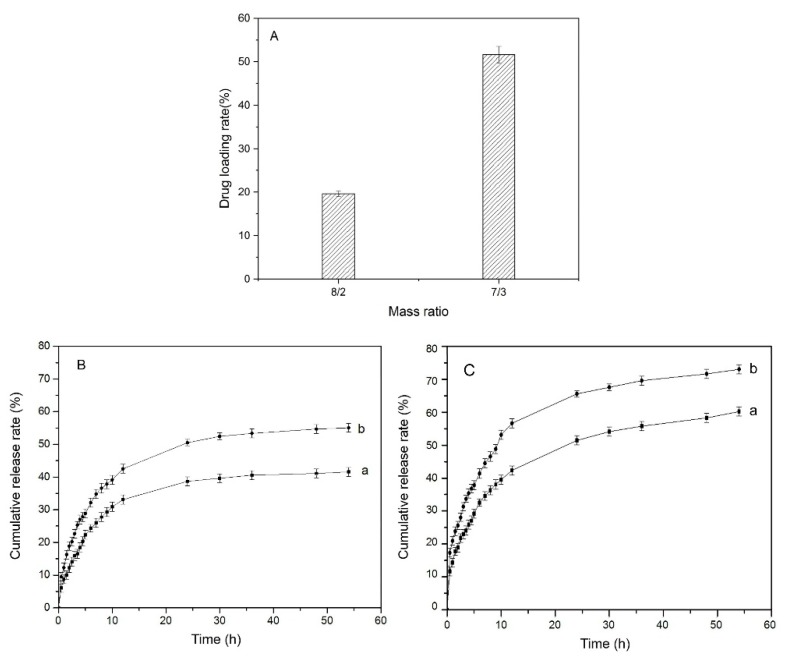
Drug loading and in vitro release rate. (**A**) Drug loading, (**B**) cumulative release rate at pH = 7.4, (**C**) cumulative release rate at pH = 2.2, (**a**) SF/CS = 8/2, (**b**) SF/CS = 7/3.

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
