# Peer review of "An Environmentally Sensitive Silk Fibroin/Chitosan Hydrogel and Its Drug Release Behaviors"

_polymers, 2019, doi:10.3390/polym11121980_

Round 1
Reviewer 1 Report
Research on the physicochemical properties of SF/CF hydrogels and future prospects for DDS. I think it is a report that provides very useful findings.
Before publishing this paper several issues should be addressed. Especially ambiguous descriptions should be clarified.
1) Figure appearance: “Figure X” instead of “Figure. X”.
2) Line 11: Tel? (optional; include country code; if there are multiple corresponding authors, add author initials)?
3) Line 24-25: The change of the gel to pH-, ion-,and MB release- stimulation should be clarified.
4) Line 35-36: It is better to have a reference.
5) Line 36-37: A description and reference should be added regarding the need for drug release using gels.
6) Line 53: Clarify what the stimulus. For example・・・ pH- and ion intensity- stimuli.
7) Line 73: Include company details・・・Sigma-Aldrich Chemical Co (Wisconsin, USA) ?
8) Line 95-97: Is the free EDC-NHS removed?
9) Line 153-155: Did you wash the hydrogels containing the MB? Or did the hydrogels load all the MB solution?
10) Line 171-172: I don't know which data led to the result and discussion. Isn't it a content to summarize in conclusuons?
11) Line 181-183: The comparison target is not clear. Compared to 9/1 and 6/4? or NSF hydrogel?
12) Figure 1B: Should clarify the mass ration used in the gel.
13) Line 191 and line 202: Figure.2A ? Isn't it “Figure 2”?
14) Line 194-197: unclear. It should be clear what is being compared with what.
15) Line 202-203: The reasons for “potential application for drug” should be clarified in references. Or you should discuss it with MB release-results.
16) Figure 2: The scale is out of order (SF/CS=9/1, 6/4). It should be corrected and clarified.
17) Line 275-281: It is better to explain that the curve of Figure7A-(a:mass rate =9/1) is different.
18) Figure 8: The vertical axis description (ESR or Equilibrium swelling rate, SR or dynamic swelling rate) should be prepared. It is better to align 8/2 (Figure 8C). Plot is small.
19) Line 311-315: Unclear. If it is a continuation of the above, it should not be broken.
20) Line 338 The release stage reached equilibrium after 35 h and then sustained release followed. : It is unlikely that sustained release followed (Figure 9B, pH 7.4).
21) Ref 45-46: The release rate varies depending on the drug. The results of research in MB or contrast in insulin release in SF gel should be compared.
22) Figure 9BC: The scale of the vertical axis should be adjusted. Or you should put it in one figure.
23) Line 355-356: It is unclear to summarize only stimulus responses. Should describe more the characteristics of different hydrogels (8/2, 7/3).
24) references: Should follow the appearance.
Author Response
Response to Reviewer 1 Comments
Point 1: Figure appearance: “Figure X” instead of “Figure. X”. 

Response 1: Modification has been completed.
Point 2: Line 11: Tel? (optional; include country code; if there are multiple corresponding authors, add author initials)?
Response 2: Telephone number has been added.
Point 3: Line 24-25: The change of the gel to pH-, ion-,and MB release- stimulation should be clarified.
Response 3: Line 25-26: With the increase of pH and ion concentration, the swelling ratios of hydrogels (SF/CS=8/2 and 7/3) decrease.
Point 4: Line 35-36: It is better to have a reference.
Response 4: Line 36: A reference (Rate- and Extent-Limiting Factors of Oral Drug Absorption: Theory and Applications) has been added to clarify the limitations of oral drug absorption.
Point 5: Line 35-36: A description and reference should be added regarding the need for drug release using gels.
Response 5: Line 38: A reference (Delivery strategies for sustained drug release in the lungs) has been added.
Point 6: Line 53: Clarify what the stimulus. For example・・・ pH- and ion intensity- stimuli.
Response 6: Line 53: In this study, a pH- and ion intensity- stimuli responsive silk fibroin/chitosan hydrogel was prepared by chemical cross-linking.
Point 7: Line 73: Include company details・・・Sigma-Aldrich Chemical Co (Wisconsin, USA) ?
Response 7: Line 74-76: EDC/NHS were purchased from Sigma-Aldrich Chemical Co. (St. Louis, USA). . The ethanol, sodium hydroxide, hydrochloric acid, phosphate buffer saline (PBS) and anhydrous sodium carbonate were all of analytical grade and purchased from Sigma-Aldrich Chemical Co. (St. Louis, USA).
Point 8: Is the free EDC-NHS removed?
Response 8: Line 98: The obtained SF/CS hydrogels were placed in deionized water for 24 h to remove residual acetic acid, ethanol and the free, unreacted EDC-NHS.
Point 9: Line 153-155: Did you wash the hydrogels containing the MB? Or did the hydrogels load all the MB solution?
Response 9: Line 155: Dry hydrogel of 40 mg was soaked in 15 mL MB solution (2 mg mL-1) to a constant weight to load drugs.
Point 10: Line 171-172: I don't know which data led to the result and discussion. Isn't it a content to summarize in conclusuons? 

Response 10: Line 171-172: In the pre-experiments, SF/CS composite hydrogel was successfully prepared.
Point 11: Line 181-183: The comparison target is not clear. Compared to 9/1 and 6/4? or NSF hydrogel?
Response 11: Line 182-184: Compared with the other composite hydrogels, hydrogels with mass ratios of 8/2 and 7/3 had better moulding performance, thus suitable for drug loading and other tests.
Point 12: Figure 1B: Should clarify the mass ration used in the gel.

Response 12: Figure 1: (B) schematic representation of SF/CS hydrogel with mass ratio of 7/3
Point 13: Line 191 and line 202: Figure.2A ? Isn't it “Figure 2”?
Response 13: Modifications have been completed.
Point 14: Line 194-197: unclear. It should be clear what is being compared with what.
Response 14: Line 195-196: When the content of CS is very low (SF/CS=10/0, 9/1 and 8/2), the pore size of these hydrogels presents larger than hydrogels (SF/CS=7/3, 6/4) .
Point 15: Line 202-203: The reasons for “potential application for drug” should be clarified in references. Or you should discuss it with MB release-results.
Response 15: Line 204: A reference (Chitosan composite hydrogels reinforced with natural clay nanotubes.) has been added.
Point 16: Figure 2: The scale is out of order (SF/CS=9/1, 6/4). It should be corrected and clarified.
Response 16: Modifications have been completed.
Point 17: Line 275-281: It is better to explain that the curve of Figure7A-(a:mass rate =9/1) is different.
Response 17: Line 275-277: The hydrogel with mass ratio of 9/1 exhibited different stress-strain curve shape due to its bad moulding performance.
Point 18: Figure 8: The vertical axis description (ESR or Equilibrium swelling rate, SR or dynamic swelling rate) should be prepared. It is better to align 8/2 (Figure 8C). Plot is small.
Response 18: Modifications have been completed.
Point 19: Line 311-315: Unclear. If it is a continuation of the above, it should not be broken.
Response 19: Line 313:Modifications have been completed.
Point 20: Line 338 The release stage reached equilibrium after 35 h and then sustained release followed. : It is unlikely that sustained release followed (Figure 9B, pH 7.4).
Response 20: Line 340: The release stage reached equilibrium after 35 h.
Point 21: Ref 45-46: The release rate varies depending on the drug. The results of research in MB or contrast in insulin release in SF gel should be compared.
Response 21: Line 343-347: In previous studies, a citric acid crosslinked NaCMC-HPMC hydrogel reached equilibrium after 12 h and release rate of methylene blue was approximately 62.75±1.40% within 4 h [47]. The pH-responsive hydrogel prepared by gelatin and bacterial cellulose reached equilibrium after 10 h, the release rate was 60% in acidic environment and 80% in normal physiological environment [48].
Point 22: Figure 9BC: The scale of the vertical axis should be adjusted. Or you should put it in one figure.
Response 22: Figure 9:Modifications have been completed.
Point 23: Line 355-356: It is unclear to summarize only stimulus responses. Should describe more the characteristics of different hydrogels (8/2, 7/3).
Response 23: Line 358-359: With the increase of pH and ion concentration, the swelling ratios of hydrogels (SF/CS=8/2 and 7/3) decrease.
Point 24: references: Should follow the appearance.
Response 24: Modifications have been completed.

Reviewer 2 Report
It is already known that silk fibroin/chitosan based composite materials have high potential to be used in biomedical applications. In the Introduction, line 50, it would be beneficial to cite also more recent reviews containing description of current state-of-the-art, e.g. Li J, Mooney DJ. Designing hydrogels for controlled drug delivery. Nature Review Materials, 2016, 1, article number 16071 or Mantha et al Smart Hydrogels in Tissue Engineering and Regenerative Medicine. Materials 2019, 12(20), 3323; https://doi.org/10.3390/ma12203323
In line 58 more recent publications can be also cited e.g. Kapoor S, Kundu SC. Silk protein-based hydrogels Promising advanced materials for biomedical applications. Acta Biomaterialia, 2016,31, 17-32
In line 100 please correct grammatical error, must be scanning instead of sanning
In lines 191 and 202 was used designation Figure. 2A but in line 204 designation Figure 2.
Author Response
Response to Reviewer 2 Comments
Point 1: It is already known that silk fibroin/chitosan based composite materials have high potential to be used in biomedical applications. In the Introduction, line 50, it would be beneficial to cite also more recent reviews containing description of current state-of-the-art, e.g. Li J, Mooney DJ. Designing hydrogels for controlled drug delivery. Nature Review Materials, 2016, 1, article number 16071 or Mantha et al Smart Hydrogels in Tissue Engineering and Regenerative Medicine. Materials 2019, 12(20), 3323; https://doi.org/10.3390/ma12203323
In line 58 more recent publications can be also cited e.g. Kapoor S, Kundu SC. Silk protein-based hydrogels Promising advanced materials for biomedical applications. Acta Biomaterialia, 2016,31, 17-32
Response 1: Some old references have been replaced.
Point 2: In line 100 please correct grammatical error, must be scanning instead of sanning
In lines 191 and 202 was used designation Figure. 2A but in line 204 designation Figure 2.
Response 2: Modifications have been completed.
